# On the effect of flexible adjustment of the p value significance threshold on the reproducibility of randomized clinical trials

**Farrokh Habibzadeh** [ORCID] *

Independent Research Consultant, Shiraz, Iran

* Farrokh.Habibzadeh@gmail.com

## Abstract

### Background

Reproducibility crisis is among major concerns of many scientists worldwide. Some researchers believe that the crisis is mostly attributed to the conventional p significance threshold value arbitrarily chosen to be 0.05 and propose to lower the cut-off to 0.005. Reducing the cut-off, although decreases the false-positive rate, is associated with an increase in false-negative rate. Recently, a flexible p significance threshold that minimizes the weighted sum of errors in statistical inference tests of hypothesis was proposed.

### Methods

The current *in silico* study was conducted to compare the error rates under different conditions assumed for the p significance threshold—0.05, 0.005, and a flexible threshold. Using a Monte Carlo simulation, the false-positive rate (when the null hypothesis was true) and false-negative rate (when the alternative hypothesis was true) were calculated in a hypothetical randomized clinical trial.

### Results

Increasing the study sample size was associated with a reduction in the false-negative rate, however, the false-positive rate occurred at a fixed value regardless of the sample size when fixed significance thresholds were used; the rate decreased, however, when the flexible threshold was employed. While employing the flexible threshold abolished the reproducibility crisis to a large extent, the method uncovered an inherent conflict in the frequentist statistical inference framework. Calculation of the flexible p significance threshold is only possible *a posteriori*, after the results are obtained. The threshold would thus be different even for replicas, which is in contradiction to the common sense.

provided the original author and source are credited.

**Data availability statement:** All relevant data are within the manuscript and its Supporting Information files.

**Funding:** The author(s) received no specific funding for this work.

**Competing interests:** The author have declared that no competing interests exist.

## Conclusions

It seems that relying on frequentist statistical inference and the p value is no longer a viable approach. Emphasis should be shifted toward alternative approaches for data analysis, Bayesian statistical methods, for example.

---

## Introduction

There is a growing belief that the results of many scientific research studies published, particularly those in the field of psychology, biology, and biomedical sciences, cannot be reproduced by other researchers. This has resulted in a crisis that undermines the credibility of the research findings obtained and presented in scientific articles [1–3]. Some scholars attribute this so-called "reproducibility crisis" to poorly designed and inappropriately conducted research studies. For instance, Altman once asserted that "[w]e need less research, better research, and research done for the right reasons" [4]. However, many investigators believe that even if well-designed studies are conducted, the reproducibility crisis would still persist because they believe that its main cause is overreliance and misuse of the p value, the end-product of frequentist statistical inference tests of hypothesis, commonly employed by scientists [5,6].

Over the past decades, the p value has increasingly been used in scientific literature. A study on more than 350 000 articles published between 1990 and 2015 that have been covered by *PubMed Central* revealed that there are, on average, nine p values in each article and that the rate of its reporting in the abstract of articles has increased from 7.3% in 1990 to 15.6% in 2014 [7].

Arbitrarily chosen by Ronald A. Fisher in 1925, the conventional p value significance threshold (PST) of 0.05 was indeed baseless [8]. Numerous *in silico* studies have shown that attaining a significant p value (conventionally, $p < 0.05$) merely by chance, is not unlikely. This implies that there would be a number of published research studies with false-positive results (*i.e.*, reporting that a treatment is effective, while it is really not) [9–11]. Therefore, it seems that the presumably relatively high PST would also contribute to the crisis. And, that is why some authors have called for reducing the PST from 0.05 to lower values. For example, Ioannidis has proposed (also supported by Benjamin, *et al*) to decrease the PST from the conventional value of 0.05 to 0.005 [12,13]. Reducing this cut-off would nonetheless cause another problem.

There is a trade-off between the false-positive (type I error) and false-negative (type II error) rates—decreasing the PST, although reduces the false-positive rate, is associated with an increase in the false-negative rate, hence, a decrease in the study power (*i.e.*, the probability that a research study will correctly identify an effect when it truly exists) [5,14]. Therefore, although lowering the PST decreases the probability of making a type I error, it may be associated with an increase in the total (type I+type II) error.

Actually, all the proposals for a smaller but fixed PST suffer from the very same problem that the conventional PST of 0.05 has faced with—there is no logic behind

choosing the values. Probably, that is why none of the proposals has so far gained universal acceptance [6]. Several scholars have proposed the importance of tailoring PST to specific research contexts [15–17]. Many of these authors believe that we need to justify PST by minimizing or balancing type I and type II errors to arrive at more scientifically valid conclusions [15–17].

In two recent articles, I have proposed a method to compute the most appropriate PST [5,6]. Using the analogy existing between the diagnostic tests with continuous results and statistical inference tests of hypothesis, in the method I proposed, the most appropriate PST is calculated in the same way as the most appropriate cut-off value for a diagnostic test is determined [5,6,18]. I defined the most appropriate PST as the value where the weighted sum of type I and type II errors in statistical inference is a minimum, just similar to how I defined the most appropriate cut-off value for a diagnostic test [5,6,18]. Given this definition, the most appropriate PST is no longer a fixed value for all studies; it depends on the study sample size, the minimum acceptable effect size, the prior probability that the alternative hypothesis ($H_1$) is correct, and the dispersion (standard deviation [SD]) of data in the study groups, among other things [5,6]. Reinterpretation of the results of 22 500 randomized clinical trials revealed that around two-thirds of the studied trials that were found "significant" based on a "p < 0.05" criterion, would have been "not significant," had the flexible PST criterion been used [5].

Although the studied trials would be considered false-positive, labeling results of a study false-positive or false-negative without prior knowledge that the null hypothesis ($H_0$) or the alternative hypothesis ($H_1$) is true, is not possible—a situation not happens in real world. The current *in silico* study, where it was possible to definitely know whether $H_0$ or $H_1$ is true, was thus conducted to determine the effect of using the different PST criteria—the conventional PST of 0.05, a lower PST of 0.005 (as proposed by Ioannidis and Benjamin) [12,13], and a flexible PST [5,6]—on the error rates in statistical inference tests of hypothesis, under different initial conditions.

## Results

For each PST criterion, the misclassification (false-positive and false-negative) rate increased as the sample size became smaller and the ratio of the SD of the treatment ($s_2$) to the placebo ($s_1$) group, $s_2/s_1$, increased (Fig 1). The false-positive (type I error) rate (Fig 1, the dark gray area) was fixed for all studied conditions when either the fixed PST criterion of 0.05 (orange column) or 0.005 (red column) was employed. The rate was different for flexible PST criterion (Fig 1, dark gray-filled blue and green columns). Depending on the situation, the false-negative (type II error) rate was variable (Fig 1, light gray filled areas). Nonetheless, for each PST criterion, the rate increased as the sample size became smaller and the $s_2/s_1$ increased (Fig 1). In most instances, in studies with smaller sample size (100 individuals or lower per arm), the prevailing error was type II error; for larger studies, type I error (Fig 1).

The weighted sum of errors (Eq. 4) followed the same pattern as the misclassification rate (Fig 2). For each scenario, the weighted sum of errors associated with the flexible threshold was a minimum (Fig 2, green column). Although the fixed threshold of 0.005 (Figs 1-4, red columns) was associated with a lower false-positive (type I error) rate as compared with the fixed conventional cut-off value of 0.05, in most instances, particularly when the sample size was smaller than 100 per study arm, it would result in an increase in the total misclassification rate (Fig 1) and the weighted sum of errors (Figs 2-4).

The pattern observed in the weighted sum of errors described above holds for other values of the prior probability that $H_1$ is true (Fig 3). Assuming a fixed $s_2/s_1$ of 0.5 and equal seriousness of type I and type II errors, the contribution of type I error (false-positive result) is lower for higher values of the prior probability of $H_1$ (Fig 3). The flexible PST criterion was associated with the least weighted sum of errors; the criterion creates a balance between the type I and type II errors (Fig 3, green column). If the variances could be considered almost equal in both study arms, the use of the conventional threshold of 0.05 was associated with higher error rates (mostly type I), when the sample size exceeded 200 per arm (Figs 1–4).

Assuming a fixed $s_2/s_1$ of 1.5, a similar pattern holds for other combinations of the prior probability that $H_1$ is true and different values for the seriousness of type II relative to type I errors (Fig 4). A reduction in the prior probability was

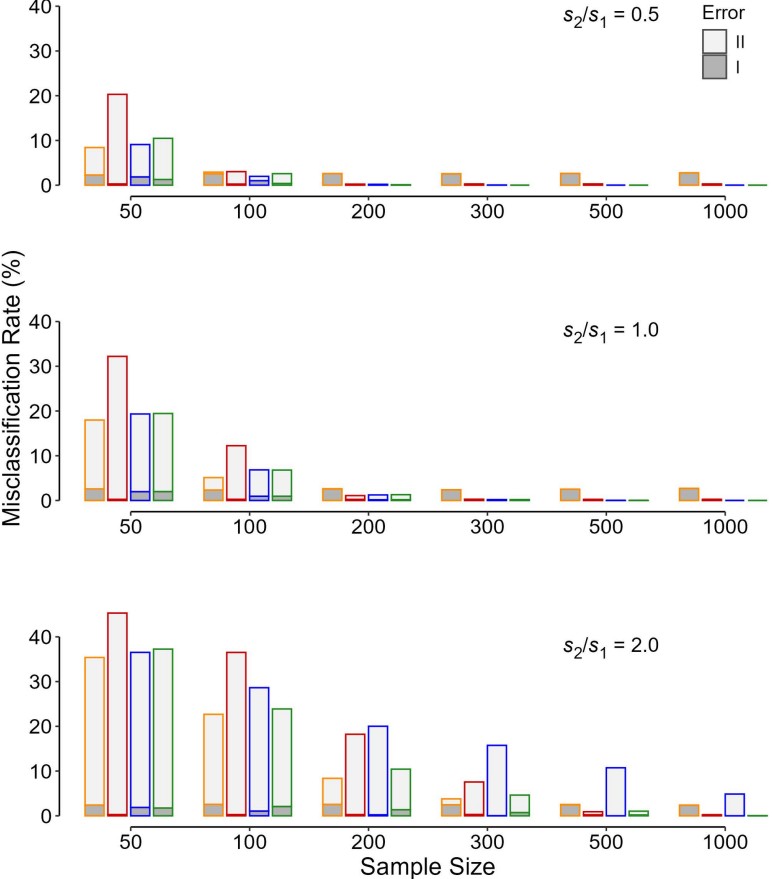

**Fig 1. Relative frequency of type I (dark gray) and type II (light gray) errors in 10 000 replicas for each initial condition of the simulation.** The values are for different sample sizes for study arm, ratios of the standard deviations, and the criteria used for determining the p value significance threshold—orange, $p < 0.05$; red, $p < 0.005$; blue, the flexible p significance threshold assuming equal variances; and green, the flexible p significance threshold for 3 variance ratios; $s_1$ and $s_2$ are the standard deviations in the placebo and treatment groups, respectively.

associated with a decrease in the rate of type II error (Fig 4). The rate increased as the seriousness of type II relative to type I error increased (Fig 4). The flexible PST criterion was associated with the least weighted sum of errors, anyway.

The most appropriate PST was highly sensitive to the $s_2/s_1$ ratio (Fig 5). As the sample size got larger, the discrepancy between the PST values calculated for each ratio also increased, so that assuming equal variances ($s_2/s_1 = 1$, Fig 5, horizontal gray lines) did not represent the most appropriate PST under various situations. Moreover, for the inherent sampling variation in replicas, the threshold substantially varied from replica to replica, even under the very same initial condition, same sample size and $s_2/s_1$ ratio (Fig 5).

The function *nleqslv* used to solve Eq. 10, successfully converged to the solution for all 1 080 000 rounds of the simulation.

## Discussion

To rely on the results of a research study, it is mandatory that the study be reproducible. Reproducibility can be considered at three levels: 1) A study should be repeatable; having enough information on the methodology used, another researcher should independently be able to replicate the same study in exactly the same way as the original researcher did. 2) The

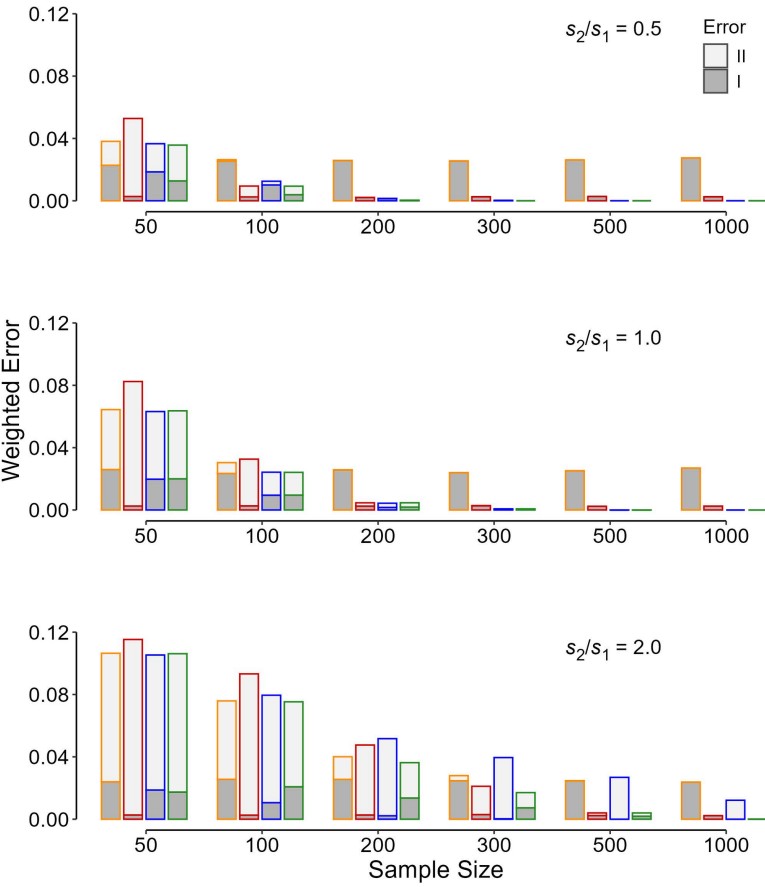

**Fig 2. The computed weighted sum of type I (dark gray) and type II (light gray) errors ( Eq. 4) in 10 000 replicas for each initial condition of the simulation.** The values are for different sample sizes for study arm, ratios of the standard deviations, and the criteria used for determining the p value significance threshold—orange, $p < 0.05$; red, $p < 0.005$; blue, the flexible p significance threshold assuming equal variances; and green, the flexible p significance threshold for 3 variance ratios; $s_1$ and $s_2$ are the standard deviations in the placebo and treatment groups, respectively.

results obtained from a study should be reproducible. The replica of a study should reproduce the same findings, taking into account the expected sampling variation. And, 3) the same conclusion should be made based on the findings obtained from the replica. In the present simulation study, the very same methodology was used for each round of the simulation (see Supplementary Materials). Therefore, repeatability was completely maintained. The presence of false-positive and false-negative results (lack of reproducibility) in the present study should thus be attributed to the lack of the second or third type of reproducibility—lack of reproducibility of the results and insufficient ability to correctly interpret the results to arrive at the same conclusion.

In the simulation, although four criteria were used to decide whether the observed difference in sample means was significant, the very same methodology was used for each criterion. Therefore, for each criterion, a given dataset would expectedly reach the same conclusion ("significant" or "not significant" difference).

Lack of reproducibility reduced with increasing sample size. The decline was more pronounced in the incidence of type II rather than type I error (Fig 1). As the sample size increases, the probability of correctly identifying an effect, if any (the study power), rises; the probability of a type II error decreases (Figs 2–4, light gray area). When a fixed PST is used, the false-positive (type I error) rate remains constant because no matter how much the fixed value is (0.05, 0.005, 0.001,

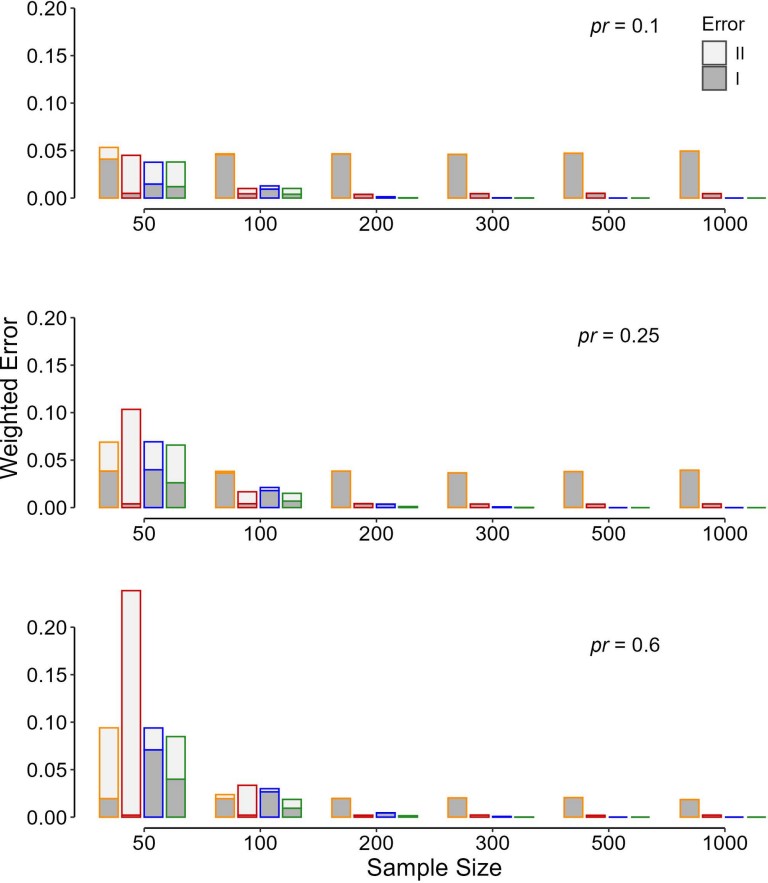

**Fig 3. The computed weighted sum of type I (dark gray) and type II (light gray) errors (** Eq. 4**) in 10 000 replicas for each initial condition of the simulation.** The values are for different sample sizes for study arm, a standard deviation ratio ($s_2/s_1$) of 0.5, a minimum effect size of interest of 0.5, and different prior probability (*pr*) values that $H_1$ is true. In this simulation, the seriousness of type I and type II errors was taken equal. The line color indicates the criteria used for determining the p value significance threshold—orange, p < 0.05; red, p < 0.005; blue, the flexible p significance threshold assuming equal variances; and green, the flexible p significance threshold.

*etc*), the rate is an inherent part of the statistical inference tests of hypothesis (Figs 1-4, dark gray area in orange and red columns).

In the era of big data and for large studies (*e.g.*, large multicenter and multinational trials) commonly conducted nowadays, the probability of type II error (false-negative results; Figs 1–4, light gray area) approaches zero. The only concern would be the false-positive (type I error) rate (Figs 1–4, dark gray areas). If a fixed PST is going to be employed, a cut-off of 0.005 is preferable to the conventional set value of 0.05 [10,12]. A cut-off of 0.001 or lower, may even be found more effective in large studies. However, this fixed value, no matter how tiny it is, will remain in all the statistical analyses. A flexible PST would work much better [5,6].

The flexible PST was associated with the minimum weighted sum of errors (Eq. 4), and it is apparently superior to fixed PSTs (Figs 2–4). However, computation of the flexible PST needs knowledge about many statistics that are only available *a posteriori* (*e.g.*, the SD of the data in the two samples), after the study was conducted [3,5,6]. Assuming equal variances in the calculation of the flexible PST was not acceptable as it did not work better than the fixed PSTs examined (Figs 1–4, blue columns). In fact, the flexible PST is highly sensitive to the variation in the SD ratio of the treatment to placebo group,

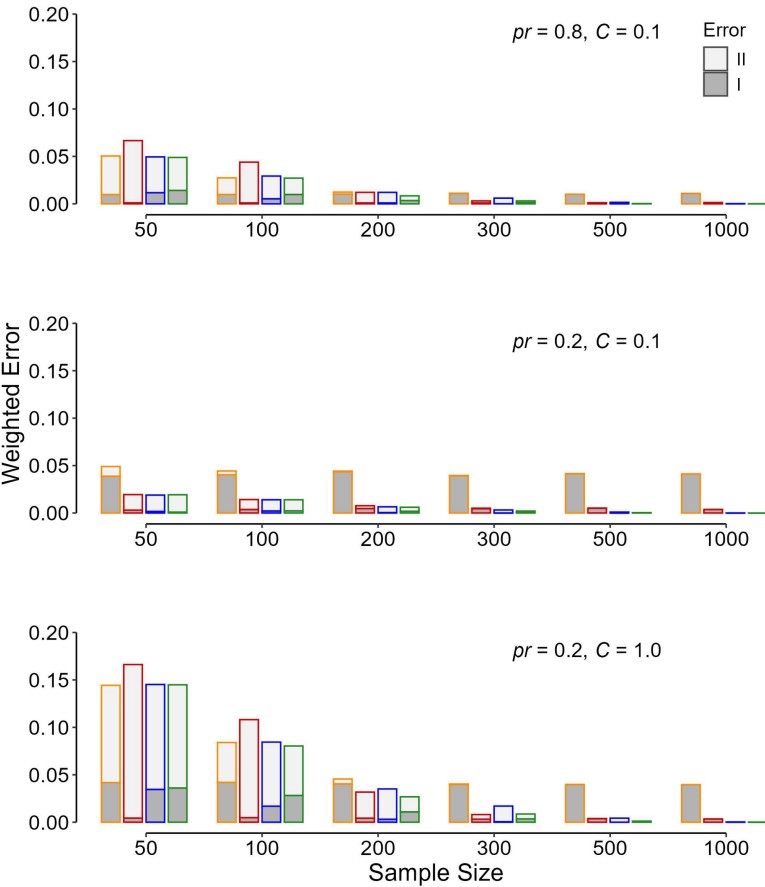

**Fig 4. The computed weighted sum of type I (dark gray) and type II (light gray) errors ( Eq. 4) in 10 000 replicas for each initial condition of the simulation.** The values are for different sample sizes for study arm, a standard deviation ratio ($s_2/s_1$) of 1.5, a minimum effect size of interest of 0.5, different prior probability (*pr*) values that $H_1$ is true, and different values for the seriousness of type II relative to type I error (*C*). The line color indicates the criteria used for determining the p value significance threshold—orange, p<0.05; red, p<0.005; blue, the flexible p significance threshold assuming equal variances; and green, the flexible p significance threshold.

$s_2/s_1$ (Fig 5), so that even if one has an estimation of the ratio *a priori*, it is not possible to calculate the threshold accurately (Fig 5)—the dispersion observed in the flexible PST, even for exactly similar initial values (sample size, variance ratio, the minimum effect size of interest, *etc*), is not trivial (Fig 5). Therefore, the flexible PST although can solve the reproducibility issue to a large extent (by minimizing the weighted sum of errors), cannot be computed *a priori*; it can only be calculated *a posteriori*.

Determining the PST *a posteriori* would leave room for parameter manipulation and p-hacking, which may introduce bias that ultimately affects the reliability and reproducibility of research findings [19]. Even for the very similar replicas of the same study, the PST would be very different for the sampling variation (Fig 5). This difference is primarily attributed to the sampling variation. This is a major problem because to avoid bias (including p-hacking), all statistical methods, including the choice of PST, should be clearly stated before the study is conducted. This underscores a salient challenge inherent to the frequentist statistical inference framework [3,5,6]. Let us illustrate this with a case study.

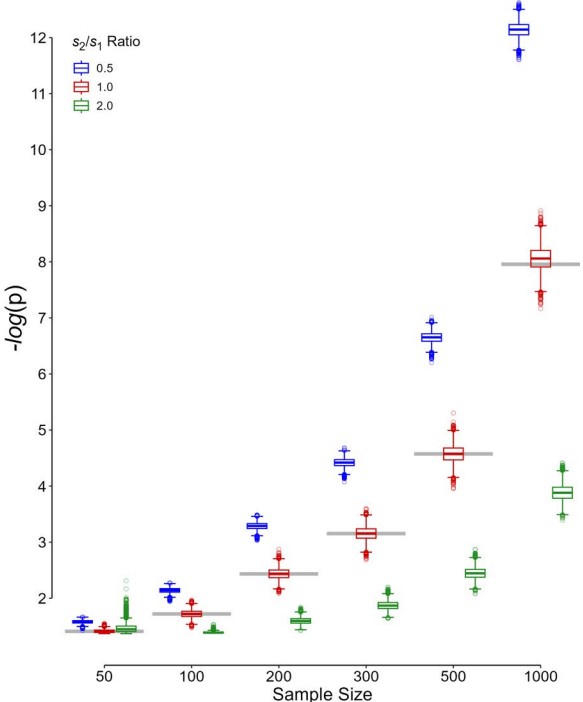

**Fig 5. Variation of the most appropriate p value significance threshold with different ratios of the standard deviations and sample sizes.** The bottom and top borders of each box indicate the 25th and 75th percentiles, respectively; the horizontal color line in the middle of each box shows the median. The lower whisker represents the smallest value within 1.5 times the interquartile range (IQR) less than the 25th percentile; the upper whisker indicates the largest data value within 1.5 times the IQR greater than the 75th percentile. Open circles represent outliers. The gray horizontal line indicates the p value significance threshold for each sample size assuming an equal variance. Note that the ordinate has a logarithmic scale; higher values correspond to lower p significance threshold. $s_1$ and $s_2$ are the standard deviations in the placebo and treatment groups, respectively.

## Case study

Suppose that in a clinical trial, 200 patients with iron-deficiency anemia were randomized into two equally sized arms; a placebo group and a treatment group. Suppose the mean hemoglobin (Hb) concentration was 11.1 (SD 2.0) g/dL in the placebo group and 13.4 (4.3) g/dL in the treatment group. Suppose that another replica of this study was conducted and the means and SDs were 11.0 (1.8) and 12.4 (4.4), respectively. These samples for two replicas were drawn from two normal distributions: the placebo group, from a distribution with a mean of 11.0 and SD of 2.0 g/dL; and the treatment group, from a distribution with a mean of 12.4 and SD of 4.0 g/dL. Also, assume that the prior probability that the drug was effective was 0.5 (50%) and that the minimum effect size of interest was 0.5 (*e.g.*, $0.5 \times SD_{placebo} = 1$ g/dL). Suppose that, as in most clinical trials in medicine, the seriousness of a type II error relative to a type I error is assumed to be 0.25.

Based on the results obtained, using the flexible criterion (minimizing Eq. 4), the most appropriate PSTs are 0.033 for the first replica and 0.025 for the second. The calculated p values are 0.001 for the first replica and 0.043 for the second [5,6]. Therefore, the observed difference in the first replica was statistically significant (because 0.001 < 0.033), whereas it was not in the second replica (because 0.043 > 0.025). The frequentist approach resulted in two different conclusions because the results are ultimately dichotomized as either retaining or rejecting a hypothesis. Bayesian methods provide a more flexible framework.

Let us do the analysis based on a Bayesian approach. It is known that:

$$Posterior\ Odds\,(H_1) = LR \times Prior\ Odds\,(H_1) \tag{1}$$

where $LR$ is the likelihood ratio [20]. The $LR$ of observing such results is 39 for the first replica and 6 for the second replica (the $LR$ for a given observed value can be calculated as the ratio of the density function [here, the Student's $t$ distribution] under $H_1$ and $H_0$ [20]). Using Eq. 1 and taking into account that the prior odds of $H_1$ is 1 (prior probability of 0.5), the posterior odds of $H_1$ are 39 and 6 for the first and second replicas, corresponding to posterior probabilities of 0.98 (98%) and 0.86 (86%), respectively. The Bayesian approach does not dichotomize the results; it just revises the posterior probability in light of the observed data. For our example, the approach says that given the observations made in each of the replicas, we can conclude that the probability that $H_1$ is true (the drug is effective) is greater than 85%.

## Numerical methods

To minimize Eq. 4, we need to solve a non-linear transcendental equation (Eq. 10), which typically has no analytical solution. Previously, I solved the problem analytically for a one-tailed problem [6], but it is not possible for a two-tailed problem. Therefore, numerical methods were used to find the solution. Fortunately, the function *nleqslv* successfully converged to a solution in all 1 080 000 rounds of the simulation. The non-linear equation to be solved (Eq. 10) was a smooth well-behaved even function. The complete success of *nleqslv* in finding the solutions may be attributed to the algorithm used by the function—Broyden's method [21]—which works very well under these conditions. However, there would be conditions that *nleqslv* cannot converge to a solution for Eq. 10.

## Conclusions

In conclusion, the evidence suggests that employing a fixed PST does not work properly; a flexible PST does. However, the concept of using a flexible PST while solving the reproducibility crisis to a large extent, in theory, does not sound like a pragmatic method, probably because of an internal conflict exists within the frequentist statistical inference paradigm. It appears that the reliance on frequentist statistical inference and the p value is no longer a viable approach [3]. Instead, emphasis should be shifted toward employing alternative approaches for data analysis. Bayesian statistical methods, while not seamless, would be considered an alternative approach to data analysis. However, it is crucial to acknowledge the limitations of the method. One significant drawback is the selection of an appropriate prior, which can be subjective and may greatly impact the results. Nonetheless, when weighing the *pros* and *cons* of Bayesian methods against frequentist approaches, Bayesian analysis appears to provide more reproducible results.

## Materials and methods

### Ethics

This *in silico* study did not involve any humans or animals or their tissue samples. Therefore, no institutional review board approval was necessary.

### Simulation

A Monte Carlo simulation was used in this study. It is a type of simulation that treats the model parameters as stochastic or random variables. It repeatedly runs the model, each time with a different randomly set of input values drawn from a set of all possible values to estimate the set of possible results (see Supplementary Materials) [22].

### Scenarios

Assume a hypothetical clinical trial was conducted to determine if a certain treatment was effective or not. Assume that the prior probability that the treatment was effective was 0.5 (*i.e.*, the probability of $H_1$), that the least effect size of interest was 0.5 (Cohen's $d$, a medium effect size) [23], and that the seriousness of making a type II error ($\beta$, stating that the treatment was not effective, while it truly was) relative to type I error ($\alpha$, stating that the treatment was effective, while it really was not) was 0.25.

The $H_0$ is that the drug has no effect, while the $H_1$ states that the drug is effective. Type I error (a false positive) could lead to widespread adoption of the ineffective drug, subjecting patients to unnecessary costs, potential side effects, and delayed access to more effective treatments. As this has serious consequences, many researchers set $a$ at a stringent level, such as 0.05 (5% chance of making this error). On the other hand, type II error (a false negative) would result in missing an opportunity to provide patients with a potentially life-saving treatment. However, $\beta$ is often set at a more lenient level, such as 0.2 (20% chance of making this error), reflecting a higher tolerance for the possibility of overlooking an effective treatment. Many investigators working in the fields of biomedical sciences believe that the maximum acceptable probability of making a type I error is 0.05; type II error, 0.2, which implies that the tolerance for making a type I error is one-fourth of that making a type II error [23–25]. The relative weighting of type I and type II errors reflects the priority often given to avoiding the approval of ineffective treatments over the rejection of effective ones. In clinical trials, this is a deliberate trade-off to protect patients from unnecessary harm while balancing the need to identify effective interventions.

The simulation was conducted for two scenarios (Table 1, see also Supplementary Materials): when $H_0$ was true (*i.e.*, there was no treatment effect), and when $H_1$ was true (*i.e.*, there was a treatment effect with an effect size of 0.5). Two samples were randomly drawn from two Gaussian distributions—the first sample for the placebo group, was taken from a distribution with a mean of zero and a SD of 1 (Table 1, steps 1 and 3); the second for the treatment group, was selected from a distribution with a mean of either zero (when $H_0$ was assumed to be true) or 0.5 (Table 1, step 2), the minimum effect size of interest (when $H_1$ was assumed to be true), and a SD of 0.5, 1.0, and 2.0. Sampling was conducted using the function *rnorm* from *R* software version 4.5.0 (*R* Project for Statistical Computing). The study assumed equal sample sizes per each arm, with the number of subjects in each arm ranging from 50 to 1000. Given the heteroscedasticity in the scenarios presented, we used Welch's *t* test to compare the means of the two groups (Table 1, step 6). The *t* statistic was calculated as follows:

$$t = \frac{m_2 - m_1}{se_\Delta} \tag{2}$$

where $m_1$ and $m_2$ represent the mean values in the placebo and the treatment groups, respectively; and $se_\Delta$, the standard error of the difference in sample means, which is:

$$se_\Delta = \sqrt{\frac{s_1^2}{n} + \frac{s_2^2}{n}} \tag{3}$$

**Table 1. Pseudocode of the simulation program (see Supplementary Materials for *R* codes S1 File).**

| | | |
|---|---|---|
| | *Loop* for different sample sizes, *n* | |
| | | *Loop* for different standard deviations of the treatment group, $s_2$ |
| | | *Loop* for 10 000 times |
| 1 | | $m_1$ = 0/* mean of the placebo group */ |
| 2 | | $m_2$ = 0 (no difference, if $H_0$ is correct) *OR* 0.5 (medium effect size, if $H_1$ is correct) |
| 3 | | Take a sample with size *n* from a normal distribution with mean $m_1$ and the standard deviation of 1, the placebo group. |
| 4 | | Take another sample with size *n* from a normal distribution with mean $m_2$ and the standard deviation of $s_2$, the treatmen t group. |
| 5 | | Compute the flexible p value significance threshold [5]. |
| 6 | | Compare the two groups with Welch's *t* test (Eqs. 2 and 3). |
| 7 | | Compare the calculated p value derived from the test with various cut-off values (0.05, 0.005, and flexible threshold) to determine if the two means are *significantly* different. |
| | | *EndLoop* |
| 8 | | Calculate the false-positive and false-negative rates and the weighted sum of errors. |
| | | *EndLoop* |
| | *EndLoop* | |

where $s_1$ and $s_2$ are the standard deviations of the placebo and treatment groups, respectively; and $n$, the sample size in each arm.

For each combination of the parameters (initial condition) studied, the simulation was run for 10 000 times (Table 1, see also Supplementary Materials). Significance of each simulated study was determined based on the p value computed, using four criteria—the conventional PST of 0.05, a lower PST of 0.005, and the flexible PST that minimized the weighted sum of errors (Table 1, step 5), as described earlier and briefly below [5]. The false-positive and false-negative rates as well as the weighted sum of errors were then computed for each scenario (Table 1, step 8).

The PST changes with the variance ratio between the two study groups. In the present study, two flexible PSTs were calculated—one assuming a variance ratio of 1; another, using the measured variance ratio (see Supplementary Materials).

### Calculation of the amount of error

The following cost function, which has been described in detail in previous studies [5,6], was used to compute the amount of weighted sum of errors ($\varepsilon$) associated with a PST of $x$ in the hypothetical studies:

$$\varepsilon\left(x\right) = Cpr\beta\left(x\right) + \left(1 - pr\right)\alpha\left(x\right) \tag{4}$$

where $C$ represents the seriousness of type II relative to type I error (assumed to be 0.25), $pr$ is the prior probability that $H_1$ is correct, and $\alpha$ and $\beta$ are the probabilities of making type I and II errors, respectively [5,6]. This is in fact equivalent to the cost function developed for the computation of the weighted number needed to misdiagnose for a diagnostic test [26]. The flexible PST was the value that minimizes the weighted sum of errors (Eq. 4) [5,6]. To minimize Eq. 4, we need to solve the following equation:

$$\frac{\partial\varepsilon\left(x\right)}{\partial x} = Cpr\frac{\partial\beta\left(x\right)}{\partial x} + \left(1 - pr\right)\frac{\partial\alpha\left(x\right)}{\partial x} = 0 \tag{5}$$

The $\alpha(x)$ and $\beta(x)$ for a two-tailed statistical test are [5]:

$$\alpha\left(x\right) = 2F_\nu\left(-|x|\right)$$
$$\beta\left(x\right) = F_\nu\left(|x| - \delta\right) - F_\nu\left(-|x| - \delta\right) \tag{6}$$

where $F_\nu$ is the cumulative distribution function of Student's $t$ distribution with a degree of freedom of $\nu$, which was calculated using the Welch-Satterthwaite equation [27,28]. The values were calculated using the function $t.test$ in $R$ (see Supplementary Materials). From Eq. 6, we have:

$$\frac{\partial\alpha\left(x\right)}{\partial x} = -2t_\nu\left(|x|\right)$$
$$\frac{\partial\beta\left(x\right)}{\partial x} = t_\nu\left(|x| - \delta\right) + t_\nu\left(|x| + \delta\right) \tag{7}$$

where $t_\nu$ is the probability density function of Student's $t$ distribution with the degree of freedom of $\nu$, which is:

$$t_\nu\left(x\right) = \frac{\Gamma\left(\frac{\nu+1}{2}\right)}{\sqrt{\pi\nu}\,\Gamma\left(\frac{\nu}{2}\right)}\left(1 + \frac{x^2}{\nu}\right)^{-(\nu+1)/2} \tag{8}$$

where $\Gamma(x)$ is the gamma function and,

$$\delta = d\frac{s_1}{se_\Delta} \tag{9}$$

Plugging in the values in Eq. 5 yields:

$$\frac{\left(1 + \frac{x^2}{\nu}\right)^{-(\nu+1)/2}}{\left(1 + \frac{(x-\delta)^2}{\nu}\right)^{-(\nu+1)/2} + \left(1 + \frac{(x+\delta)^2}{\nu}\right)^{-(\nu+1)/2}} = \frac{C\,pr}{2\,(1-pr)}$$

(10)

For most values of the parameters, this non-linear transcendental equation has no analytical solution. Therefore, I used the function *nleqslv* from an *R* package of the same name [29], to solve Eq. 10 numerically. By default, the function uses Broyden's method (a quasi-Newton method) to solve the equation [21]. In numerical methods, the function used for optimization sometimes fails to converge to the root of interest. The convergence of the optimization was verified for each estimate by examining the convergence flag returned by *nleqslv* (see Supplementary Materials).

### Types of error

A type I error can only occur when the $H_0$ is true; a type II, when $H_1$. When the $H_0$ was true, the false-positive (type I error) rate was calculated as the proportion of simulated studies (out of 10 000 replicas) that were found "significant" (*i.e.*, the calculated p value was less than the PST specified by each criterion). Similarly, when the $H_1$ was true, the false-negative (type II error) rate was computed as the proportion of simulated studies that were found "not significant" (*i.e.*, the calculated p value was equal to or greater than the PST for each criterion). The amounts of error and weighted sum of errors (Eq. 4) were calculated for each scenario and for each of the four PST criteria used (Table 1, step 8).

### Supporting information

**S1 File. Simulation codes in *R*.**
(PDF)

### Author contributions

**Conceptualization:** Farrokh Habibzadeh.

**Data curation:** Farrokh Habibzadeh.

**Formal analysis:** Farrokh Habibzadeh.

**Investigation:** Farrokh Habibzadeh.

**Methodology:** Farrokh Habibzadeh.

**Project administration:** Farrokh Habibzadeh.

**Resources:** Farrokh Habibzadeh.

**Software:** Farrokh Habibzadeh.

**Supervision:** Farrokh Habibzadeh.

**Validation:** Farrokh Habibzadeh.

**Visualization:** Farrokh Habibzadeh.

**Writing – original draft:** Farrokh Habibzadeh.

**Writing – review & editing:** Farrokh Habibzadeh.

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
