## [Decision Letter · Decision Letter 0]

19 Mar 2025

Dear Dr. Habibzadeh,

Thank you for submitting your manuscript to PLOS ONE. After careful consideration, we feel that it has merit but does not fully meet PLOS ONE’s publication criteria as it currently stands. Therefore, we invite you to submit a revised version of the manuscript that addresses the points raised during the review process.

We look forward to receiving your revised manuscript.

Kind regards,

Christine E. King, PhD

Academic Editor

PLOS ONE

Journal Requirements:

2. Please note that PLOS ONE has specific guidelines on code sharing for submissions in which author-generated code underpins the findings in the manuscript. In these cases, we expect all author-generated code to be made available without restrictions upon publication of the work. 

Please review our guidelines at https://journals.plos.org/plosone/s/materials-and-software-sharing#loc-sharing-code and ensure that your code is shared in a way that follows best practice and facilitates reproducibility and reuse.

Reviewers' comments:

Reviewer's Responses to Questions

**Comments to the Author**

1. Is the manuscript technically sound, and do the data support the conclusions?

Reviewer #1: Partly

Reviewer #2: Yes

2. Has the statistical analysis been performed appropriately and rigorously?

Reviewer #1: Yes

Reviewer #2: Yes

3. Have the authors made all data underlying the findings in their manuscript fully available?

Reviewer #1: Yes

Reviewer #2: Yes

4. Is the manuscript presented in an intelligible fashion and written in standard English?

Reviewer #1: Yes

Reviewer #2: Yes

Reviewer #1: The computer simulation presented here is based on an interesting and sensible premise: It might be sensible to flexibly adjust the alpha level (or, “PST”, in the author’s terminology) employed in statistical significance testing in order to optimize the risks of type-1 and type-2 errors. I find most of the manuscript clear and the basic premise of the simulations convincing. However, I think that a number of issues need to be addressed.

1) I find the title unhelpful. There are many ways to determine if a replication was successful. This does not need to rely on statistical significance. I would find title that focusses on flexible adjustment of PST much more helpful.

2) The introduction is somewhat myopic. Previous discussions of the topic by other authors should be acknowledged (e.g., https://daniellakens.blogspot.com/2019/05/justifying-your-alpha-by-minimizing-or.html and references therein).

3) It would be helpful to mention early on that this paper focusses on dvs. Similarly, the important concept of weighting type-1 vs. type-2 errors (i.e., one type of error might be considered more harmful than the other) should be appropriately introduced in the introduction. An example might be helpful.

4) The introduction of unequal variances in the two simulated study arms puzzled me. What is the motivation behind it? Is this meant as a device to manipulate the true effect size (ES)? Manipulation of the ES via the difference in the population means would be much easier to understand (i.e., the difference in population means equals d when the true variance for both populations remains fixed at 1). Also, with unequal variances researchers might not rely on the t-test for analysis as is assumed in the simulation.

5) I find the data space explored by the simulation overly narrow. It would be good to systematically manipulate not just N and ES but also the relative weight given to type-1 and type-2 errors and the base rate of these errors. Dealing with results from a design with that many factors (including their interactions, which are currently neglected) is challenging, but you might find this approach helpful https://journals.plos.org/plosone/article?id=10.1371/journal.pone.0262809.

6) I don’t understand how the misclassification rate can exceed 52.5% for alpha = .05 (see Figure 1). Simulations with H0 being true lead to 2.5% misclassifications (alpha x 50% H0 base rate). In the most extreme case that statistical power = zero, simulations with H1 being true add another 50% misclassifications (beta x 50% H1 base rate). Both add to just 52.5%. I therefore wonder if an error crept into the simulation.

7) I did not understand Figure 3 and the related arguments in the discussion. Without further arguments, I don’t find it problematic when determination of the optimal PST relies on observed sample statistics.

8) In a surprising swerve, the discussion argues of the superiority of Bayes over frequentist statistics. I find that unhelpful because the simulations allow no such comparison.

Johannes Hönekopp

Reviewer #2: Dear Authors,

Thank you for submitting your manuscript, On the Reproducibility of Randomized Clinical Trials, to PLOS ONE. Your work addresses a critical issue in modern scientific research and provides valuable insights into the limitations of frequentist statistical frameworks. Below, I outline my evaluation and suggestions for improvement to strengthen the manuscript further.

Your Monte Carlo simulation study is well-designed and rigorously executed, offering a compelling comparison of fixed vs. flexible p-value thresholds. The conclusions are largely supported by the data. However, the discussion of Bayesian methods as an alternative approach remains underdeveloped. Please provide concrete examples or references illustrating how Bayesian methods (e.g., posterior probability calculations, Bayes factors) might resolve the conflicts inherent in frequentist frameworks, and discuss how Bayesian approaches address reproducibility challenges in practical terms (e.g., prior incorporation, decision thresholds).

The statistical methodology is robust, please clarify why specific parameters (e.g., prior probability

pr=0.5, effect size = 0.5) were chosen. A sensitivity analysis exploring how varying these parameters impacts the flexible threshold’s performance would strengthen the study’s applicability.

Finally, please highlight practical challenges of implementing flexible thresholds (e.g., a posteriori calculation, variability across replicas) in the discussion section.

This study makes a meaningful contribution to the discourse on reproducibility and statistical inference. Thank you for the opportunity to review your work.

Best regards,

**Do you want your identity to be public for this peer review?** For information about this choice, including consent withdrawal, please see our Privacy Policy

Reviewer #1: **Yes: ** Johannes Hönekopp

Reviewer #2: **Yes: ** Angel A. García O'Diana

---

## [Author Response · Author response to Decision Letter 1]

31 Mar 2025

Dear Editor,

I hope this message finds you well. I would like to sincerely thank you for sharing the insightful and constructive comments from the respected Editors and distinguished reviewers. I have carefully revised the manuscript in response to their valuable feedback. I believe the revised version of the manuscript has been significantly improved, and I hope you find it suitable for publication in PLoS One.

Thank you once again for your time and consideration.

Best Regards,

F. Habibzadeh, MD

Reviewer #1:

The computer simulation presented here is based on an interesting and sensible premise: It might be sensible to flexibly adjust the alpha level (or, “PST”, in the author’s terminology) employed in statistical significance testing in order to optimize the risks of type-1 and type-2 errors. I find most of the manuscript clear and the basic premise of the simulations convincing. However, I think that a number of issues need to be addressed.

1) I find the title unhelpful. There are many ways to determine if a replication was successful. This does not need to rely on statistical significance. I would find title that focusses on flexible adjustment of PST much more helpful.

Answer: Thank you very much for your good suggestion. The title of the manuscript has been changed to “On the Effect of Flexible Adjustment of the p Value Significance Threshold on the Reproducibility of Randomized Clinical Trials.”

2) The introduction is somewhat myopic. Previous discussions of the topic by other authors should be acknowledged (e.g., https://daniellakens.blogspot.com/2019/05/justifying-your-alpha-by-minimizing-or.html and references therein).

Answer: Thank you for your excellent suggestion. I have cited a few peer-reviewed references taken from the blog site you mentioned. I have also revised the Introduction to acknowledge these contributions and provide a broader context for the use of flexible significance thresholds.

3) It would be helpful to mention early on that this paper focusses on dvs. Similarly, the important concept of weighting type-1 vs. type-2 errors (i.e., one type of error might be considered more harmful than the other) should be appropriately introduced in the introduction. An example might be helpful.

Answer: Thank you for your valuable suggestion. I have revised the Methos section where I described the relative weight of type I and type II errors and elaborated on the rationale for choosing the weight of 0.25 in clinical trials. This weight is common in medical research.

4) The introduction of unequal variances in the two simulated study arms puzzled me. What is the motivation behind it? Is this meant as a device to manipulate the true effect size (ES)? Manipulation of the ES via the difference in the population means would be much easier to understand (i.e., the difference in population means equals d when the true variance for both populations remains fixed at 1). Also, with unequal variances researchers might not rely on the t-test for analysis as is assumed in the simulation.

Answer: Thank you for raising this important point. I’d like to clarify that in the simulation, the SD in the untreated group was assumed to be 1, while the SD in the treatment group assumed values of 0.5, 1, and 2. Many drugs would affect the dispersion of data in the treatment group. The introduction of unequal variances enables us to examine how the simulation behaves when the variances are not equal, which is a common real-world scenario in biomedical research. As the effect size (ES) is defined as the quotient of the minimum expected difference (d) and the SD of the untreated group (which is set to 1), the variation in the SD of the treatment group does not directly affect the ES. Regarding the use of the Student’s t test in the presence of unequal variances, the pooled estimate of the variance was used in the calculations.

5) I find the data space explored by the simulation overly narrow. It would be good to systematically manipulate not just N and ES but also the relative weight given to type-1 and type-2 errors and the base rate of these errors. Dealing with results from a design with that many factors (including their interactions, which are currently neglected) is challenging, but you might find this approach helpful https://journals.plos.org/plosone/article?id=10.1371/journal.pone.0262809.

Answer: Thank you for your valuable suggestion. I added another Figure describing the weighted error under different initial conditions. I also added a Case study to illustrate the issue. More information has already been published (see https://doi.org/10.1371/journal.pone.0305575 and https://doi.org/10.1186/s12967-023-04827-8 [References 5 and 6 of the manuscript]). To avoid duplicate publication, I do not present these published results in the current manuscript.

6) I don’t understand how the misclassification rate can exceed 52.5% for alpha = .05 (see Figure 1). Simulations with H0 being true lead to 2.5% misclassifications (alpha x 50% H0 base rate). In the most extreme case that statistical power = zero, simulations with H1 being true add another 50% misclassifications (beta x 50% H1 base rate). Both add to just 52.5%. I therefore wonder if an error crept into the simulation.

Answer: Thank you very much for pointing out this mistake. You’re entirely right. In Figure 1, in the calculation of the expected misclassification rate, I forgot to multiply the error rates by their probabilities. Figure 1 was corrected and regenerated.

7) I did not understand Figure 3 and the related arguments in the discussion. Without further arguments, I don’t find it problematic when determination of the optimal PST relies on observed sample statistics.

Answer: Thank you for your question. Figure 4 (3 in the old version) shows that even with exactly given initial conditions, the most appropriate PST values from the flexible criterion are significantly variable just for sampling variation. We cannot even compute an acceptable a priori estimate of the PST value. I find this problematic because all statistical methods, including the choice of PST, should be stated before the study is conducted (to avoid bias). Determining the PST a posteriori would leave room for parameter manipulation and p-hacking.

8) In a surprising swerve, the discussion argues of the superiority of Bayes over frequentist statistics. I find that unhelpful because the simulations allow no such comparison.

Answer: Thank you very much for your comment. I have tried to clarify the issue through a Case study and discussion. I am not saying that Bayesian analysis is flawless. I just mention it as an example of alternative ways to analyze data.

 

Reviewer #2:

Thank you for submitting your manuscript, On the Reproducibility of Randomized Clinical Trials, to PLOS ONE. Your work addresses a critical issue in modern scientific research and provides valuable insights into the limitations of frequentist statistical frameworks. Below, I outline my evaluation and suggestions for improvement to strengthen the manuscript further.

1) Your Monte Carlo simulation study is well-designed and rigorously executed, offering a compelling comparison of fixed vs. flexible p-value thresholds. The conclusions are largely supported by the data. However, the discussion of Bayesian methods as an alternative approach remains underdeveloped. Please provide concrete examples or references illustrating how Bayesian methods (e.g., posterior probability calculations, Bayes factors) might resolve the conflicts inherent in frequentist frameworks, and discuss how Bayesian approaches address reproducibility challenges in practical terms (e.g., prior incorporation, decision thresholds).

Answer: Thank you very much for your fruitful comment and suggestion. I have elaborated on the issue and provided a Case study to further clarify the topic. Please note that I am not saying that Bayesian analysis is flawless. I just mention it as an example of alternative ways to analyze data.

2) The statistical methodology is robust, please clarify why specific parameters (e.g., prior probability pr=0.5, effect size = 0.5) were chosen. A sensitivity analysis exploring how varying these parameters impacts the flexible threshold’s performance would strengthen the study’s applicability.

Answer: Thank you very much for raising this important question. The initial conditions (pr of 0.5, seriousness of type II relative to type I error of 0.25, and a minimum effect size of interest of 0.5) are common assumptions made in randomized clinical trials in medicine. I have used other initial conditions and presented the results in another Figure (Fig. 3 in the revised version). More detailed results have already been presented in other studies (see https://doi.org/10.1371/journal.pone.0305575 and https://doi.org/10.1186/s12967-023-04827-8 [References 5 and 6 of the manuscript]). To avoid duplicate publication, I do not present these published results in the current manuscript.

3) Finally, please highlight practical challenges of implementing flexible thresholds (e.g., a posteriori calculation, variability across replicas) in the discussion section.

Answer: Thank you very much for your excellent suggestion. I have tried to explain the issue. One major problem is that the flexible PST cannot be calculated a priori, which would introduce bias. To avoid bias, all statistical methods, including the PST, should be stated before the study is conducted. Determining the PST a posteriori would leave room for parameter manipulation and p-hacking.

This study makes a meaningful contribution to the discourse on reproducibility and statistical inference. Thank you for the opportunity to review your work.

Answer: Thank you very much for your kind words.

---

## [Decision Letter · Decision Letter 1]

6 May 2025

Dear Dr. Habibzadeh,

Thank you for submitting your manuscript to PLOS ONE. After careful consideration, we feel that it has merit but does not fully meet PLOS ONE’s publication criteria as it currently stands. Therefore, we invite you to submit a revised version of the manuscript that addresses the points raised during the review process.

1) The use of fixed prior probabilities (pr = 0.5) and seriousness ratios (C = 0.25) needs to be tested through sensitivity analysis. Robustness under varied assumptions would strengthen the manuscript’s generalizability.

2) The assumption of pooled variance in t-testing under heteroscedasticity requires stronger justification or a robustness check via Welch’s t-test.

3) The manuscript should comment on convergence diagnostics for the optimization routine (optim()), since flexible PST values are highly sensitive to this step.

We look forward to receiving your revised manuscript.

Kind regards,

Christine E. King, PhD

Academic Editor

PLOS ONE

Journal Requirements:

Reviewers' comments:

Reviewer's Responses to Questions

**Comments to the Author**

Reviewer #1: All comments have been addressed

Reviewer #2: All comments have been addressed

2. Is the manuscript technically sound, and do the data support the conclusions?

Reviewer #1: Yes

Reviewer #2: Yes

3. Has the statistical analysis been performed appropriately and rigorously?

Reviewer #1: Yes

Reviewer #2: Yes

4. Have the authors made all data underlying the findings in their manuscript fully available?

Reviewer #1: Yes

Reviewer #2: Yes

5. Is the manuscript presented in an intelligible fashion and written in standard English?

Reviewer #1: Yes

Reviewer #2: Yes

Reviewer #1: (No Response)

Reviewer #2: Dear Dr. Habibzadeh,

Thank you for your thoughtful and well-executed manuscript on the effect of flexible adjustment of the p-value significance threshold on the reproducibility of randomized clinical trials. Your approach offers a valuable contribution to the ongoing discussion surrounding statistical inference and reproducibility, particularly by proposing a flexible significance threshold based on minimizing a weighted sum of Type I and Type II errors.

While your manuscript is methodologically robust and generally well-presented, I would like to offer the following comments and suggestions to strengthen the manuscript further prior to publication:

Sensitivity Analysis of Prior Probability and Error Seriousness Coefficient

The current simulations assume a fixed prior probability (pr = 0.5) and a seriousness ratio (C = 0.25). While these values may reflect common scenarios in clinical research, their generalizability across fields is limited.

➤ I recommend adding a sensitivity analysis varying both pr (e.g., 0.2, 0.8) and C (e.g., 0.1, 1.0). This would enhance the robustness of your conclusions and demonstrate the flexibility and limitations of the proposed method under different decision-theoretic contexts.

Use of Pooled Variance under Heteroscedasticity

The simulations employ Student’s t-test with pooled variance, even in scenarios where variances between groups are unequal (s2/s1 ≠ 1).

➤ Please justify this methodological choice more clearly. Alternatively, consider including results using Welch’s t-test as a robustness check. This would provide greater assurance that conclusions drawn under the assumption of pooled variance are not overly sensitive to violations of homoscedasticity.

Convergence Validation of the Optimization Routine (optim)

Your method relies heavily on the optim() function to estimate flexible thresholds by minimizing a cost function. While this is an appropriate approach, the manuscript does not discuss whether convergence was reliably achieved or how non-convergence was handled.

➤ I suggest including either a brief description of the convergence diagnostics performed (e.g., convergence rate, error handling), or augmenting the code with a validation step to report non-converged iterations.

These refinements will bolster the scientific rigor and transparency of your manuscript, improving its reproducibility and interpretability for a broad audience.

Kind regards,

**Do you want your identity to be public for this peer review?** For information about this choice, including consent withdrawal, please see our Privacy Policy

Reviewer #1: No

Reviewer #2: **Yes: ** Angel Alfonso García O'Diana

---

## [Author Response · Author response to Decision Letter 2]

10 May 2025

Dear Editor,

I would like to sincerely thank you for sharing the insightful and constructive comments from the second reviewer. I have carefully revised the manuscript in response to his valuable feedback. I believe the revised version of the manuscript has been significantly improved, and I hope you find it suitable for publication in PLoS One.

Thank you once again for your time and consideration.

Best Regards,

F. Habibzadeh, MD

Reviewer #2:

Thank you for your thoughtful and well-executed manuscript on the effect of flexible adjustment of the p-value significance threshold on the reproducibility of randomized clinical trials. Your approach offers a valuable contribution to the ongoing discussion surrounding statistical inference and reproducibility, particularly by proposing a flexible significance threshold based on minimizing a weighted sum of Type I and Type II errors.

While your manuscript is methodologically robust and generally well-presented, I would like to offer the following comments and suggestions to strengthen the manuscript further prior to publication:

Sensitivity Analysis of Prior Probability and Error Seriousness Coefficient

The current simulations assume a fixed prior probability (pr = 0.5) and a seriousness ratio (C = 0.25). While these values may reflect common scenarios in clinical research, their generalizability across fields is limited.

➤ I recommend adding a sensitivity analysis varying both pr (e.g., 0.2, 0.8) and C (e.g., 0.1, 1.0). This would enhance the robustness of your conclusions and demonstrate the flexibility and limitations of the proposed method under different decision-theoretic contexts.

Answer: Thanks for your great suggestion. I have added another Figure to show the results of different pr and C values.

Use of Pooled Variance under Heteroscedasticity

The simulations employ Student’s t-test with pooled variance, even in scenarios where variances between groups are unequal (s2/s1 ≠ 1).

➤ Please justify this methodological choice more clearly. Alternatively, consider including results using Welch’s t-test as a robustness check. This would provide greater assurance that conclusions drawn under the assumption of pooled variance are not overly sensitive to violations of homoscedasticity.

Answer: Thank you very much for raising this important issue. I have changed the codes and used the Welch’s t test. The Figures and results have been updated.

Convergence Validation of the Optimization Routine (optim)

Your method relies heavily on the optim() function to estimate flexible thresholds by minimizing a cost function. While this is an appropriate approach, the manuscript does not discuss whether convergence was reliably achieved or how non-convergence was handled.

➤ I suggest including either a brief description of the convergence diagnostics performed (e.g., convergence rate, error handling), or augmenting the code with a validation step to report non-converged iterations.

Answer: Thank you for raising this important issue. I changed the methodology, tried to solve the problem analytically, and finally changed the R codes. I checked the rate of convergence for each scenario. Fortunately, the optimization function used in the revised version (nleqslv) converged in all cases.

---

## [Editor Report · Decision Letter 2]

21 May 2025

On the effect of flexible adjustment of the p value significance threshold on the reproducibility of randomized clinical trials

PONE-D-24-50574R2

Dear Dr. Habibzadeh,

We’re pleased to inform you that your manuscript has been judged scientifically suitable for publication and will be formally accepted for publication once it meets all outstanding technical requirements.

Kind regards,

Christine E. King, PhD

Academic Editor

PLOS ONE
---

## [Editor Report · Acceptance letter]

PONE-D-24-50574R2

PLOS ONE

Dear Dr. Habibzadeh,

I'm pleased to inform you that your manuscript has been deemed suitable for publication in PLOS ONE. Congratulations! Your manuscript is now being handed over to our production team.

Kind regards,

on behalf of

Dr. Christine E. King

Academic Editor

PLOS ONE